# Learning Visible Connectivity Dynamics for Cloth Smoothing

**Xingyu Lin**\*
Robotics Institute
Carnegie Mellon University
xlin3@andrew.cmu.edu

**Yufei Wang**\*
Robotics Institute
Carnegie Mellon University
yufeiw2@andrew.cmu.edu

**Zixuan Huang**
Robotics Institute
Carnegie Mellon University
zixuanhu@andrew.cmu.edu

**David Held**
Robotics Institute
Carnegie Mellon University
dheld@andrew.cmu.edu

**Abstract:** Robotic manipulation of cloth remains challenging due to the complex dynamics of cloth, lack of a low-dimensional state representation, and self-occlusions. In contrast to previous model-based approaches that learn a pixel-based dynamics model or a compressed latent vector dynamics, we propose to learn a particle-based dynamics model from a partial point cloud observation. To overcome the challenges of partial observability, we infer which visible points are connected on the underlying cloth mesh. We then learn a dynamics model over this visible connectivity graph. Compared to previous learning-based approaches, our model poses strong inductive bias with its particle based representation for learning the underlying cloth physics; it can generalize to cloths with novel shapes; it is invariant to visual features; and the predictions can be more easily visualized. We show that our method greatly outperforms previous state-of-the-art model-based and model-free reinforcement learning methods in simulation. Furthermore, we demonstrate zero-shot sim-to-real transfer where we deploy the model trained in simulation on a Franka arm and show that the model can successfully smooth cloths of different materials, geometries and colors from crumpled configurations. Videos can be found in the supplement and on our anonymous project website.[1]

**Keywords:** Deformable Object Manipulation, Dynamics Modeling

## 1 Introduction

Robotic manipulation of cloth has wide applications across both industrial and domestic tasks such as laundry folding and bed making. However, cloth manipulation remains challenging for robotics due to the complex cloth dynamics. Further, like most deformable objects, cloth cannot be easily described by low-dimensional state representations when placed in arbitrary configurations. Self-occlusions make state estimation especially difficult when the cloth is crumpled.

One approach to cloth manipulation explored by previous work, which we also adopt, is to learn a cloth dynamics model and then use the model for planning to determine the robot actions. However, given that a crumpled cloth has many self-occlusions and complex dynamics, it is unclear how to choose the appropriate state representation. One possible state representation is to use a mesh model of the entire cloth [1]. However, fitting a full mesh model to an arbitrary crumpled cloth configuration is difficult. Recent work have approached fabric manipulation by either compressing the cloth representation into a fixed-size latent vector [2, 3, 4] or directly learning a visual dynamics model in pixel space [5]. However, these representations do not enforce any inductive bias of the cloth physics, leading to suboptimal performance and generalization.

---

\* Equal contribution, order by dice rolling.
[1] https://sites.google.com/view/vcd-cloth

5th Conference on Robot Learning (CoRL 2021), London, UK.

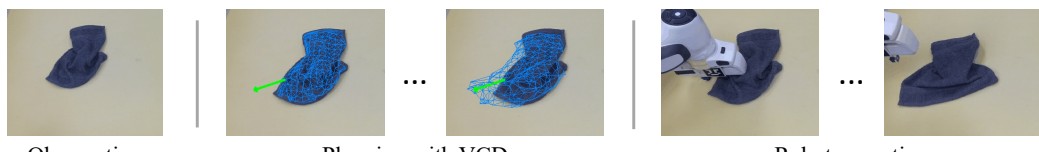

| Observation | Planning with VCD | Robot execution |

**Figure 1:** Cloth smoothing by planning using a dynamics model with a visible connectivity graph.

In contrast to a pixel-based or latent dynamics model, particle-based models have recently been shown to be able to learn dynamics for fluid and plastics [6, 7, 8]. A particle-based dynamics representation has the following benefits: first, it captures the inductive bias of the underlying physics, since real-world objects are composed of underlying atoms that can be modeled on the micro-level by particles. Second, we can incorporate inductive bias by directly applying the effect of the robot gripper on the particle being grasped (though the effect on the other particles must still be inferred). Last, particle-based models are invariant to visual features such as object colors or patterns. As such, in this paper we aim to learn a particle-based dynamics model for cloth. However, the challenges in applying the particle-based model to cloth are that we cannot directly observe the underlying particles composing the cloth nor their mesh connections. The problem is made even more challenging due to the partial observability of the cloth from self-occlusions when it is in a crumpled configuration.

Our insight into this problem is that, rather than fitting a mesh model to the observation, we should learn the *visible connectivity dynamics (VCD):* a dynamics model based on the connectivity structure of the visible portion of the cloth. To do so, we first learn to estimate the *visible connectivity graph*: we estimate which points in the point cloud observation are connected in the underlying cloth mesh (see Figure 1). Estimating the mesh connectivity of the observation is a simplification of the problem of fitting a single full mesh model of the entire cloth to the observation; however, it is significantly easier to learn, since we do not need to find a globally consistent explanation of the observation which requires reasoning about occlusion; to estimate the mesh connectivity of the observation, we only need to consider the visible local cloth structure. While the graph is constructed only based on the visible points, we show that the dynamics model can be trained to be robust to partial observation.

In this work, we focus on the task of smoothing a piece of cloth from a crumpled configuration. We propose a method that infers the observable particles and their connections from the point cloud, learns a visible connectivity dynamics model for the observable portion of the cloth, and uses it for planning to smooth the cloth. We show that for smoothing, planning with a visible connectivity dynamics model greatly outperforms state-of-the-art model-based and model-free reinforcement learning methods that use a fixed-size latent vector representation or learn a pixel-based visual dynamics model. We then demonstrate zero-shot sim-to-real transfer where we deploy the model trained in simulation on a Franka arm and show that the learned model can successfully smooth cloths of different materials, geometries, and colors from crumpled configurations.

## 2  Related Work

**Vision-based Cloth Manipulation:**   Some papers on cloth manipulation assume that the cloth is already lying flat on the table [9, 10, 11]. If the cloth starts in an unknown configuration, then one approach is to perform a sequence of actions that are designed to move the cloth into a set of known configurations from which perception can be performed more easily [12, 13, 14]. For example, the robot might first grasp the cloth by an arbitrary point and raise it into the air; it can then detect the lowest point, either while the cloth is held in the air [12, 15, 16, 17, 18, 19] or after throwing the cloth on the table [14]. By constraining the cloth to this configuration set, the task of perceiving the cloth or fitting a mesh model [1] is greatly simplified. However, these funneling actions are usually scripted and are not generalizable to different cloth shapes or configurations. In contrast, our work aims to enable a robot to interact with cloth from arbitrary configurations and shapes.

Other early works designed vision systems for detecting cloth features that can be used for downstream tasks, such as a Harris Corner Detector [20] or a wrinkle-detector [21]. More examples of such approaches are described in [1]. However, these approaches require a task-specific manual design of vision features and are typically not robust to different variations of the cloth configuration.

**Policy Learning for Cloth Manipulation:** Recently, there have been a number of learning based approaches to cloth folding and smoothing. One approach is to learn a policy to achieve a given manipulation task. Some papers approach this using learning from demonstration. The demonstrations can be obtained using a heuristic expert [22] or a scripted sequence of actions based on cloth descriptors [23]. Another approach to policy learning is model-free reinforcement learning (RL), which has been applied to cloth manipulation [4, 3, 24]. However, policy learning approaches often lack the ability to generalize to novel situations; this is especially problematic for cloth manipulation in which the cloth can be in a wide variety of crumpled configurations. We compare our method to a state-of-the-art policy learning approach [3] and show greatly improved performance.

**Model-based RL for Cloth Manipulation:** Model-based RL methods learn a dynamics model and then use it for planning. Model-based reinforcement learning methods have many benefits such as sample efficiency, interpretability, and generalizability to multiple tasks. Previous works have tried to learn a pixel-based dynamics model that directly predict the future cloth images after an action is applied [25, 5]. However, learning a visual model for image prediction is difficult and the predicted images are usually blurry, unable to capture the details of the cloth. Another approach is to represent the cloth with a fixed-size latent vector representation [2] and to plan in that latent space. However, cloth has an intrinsic high dimension state representation; thus, such compressed representations typically lose the fine-grained details of their environment and are unsuitable for capturing the low-level details of the cloth's shape, such as folds or wrinkles, which can be important for folding or other manipulation tasks. Our method also falls into the model-based RL category; unlike previous works, we learn a particle based dynamics model [6, 7], which can better capture the cloth dynamics due to the inductive bias of the particle representation. Additionally, the particle representation is invariant to visual features and enables easier sim-to-real transfer.

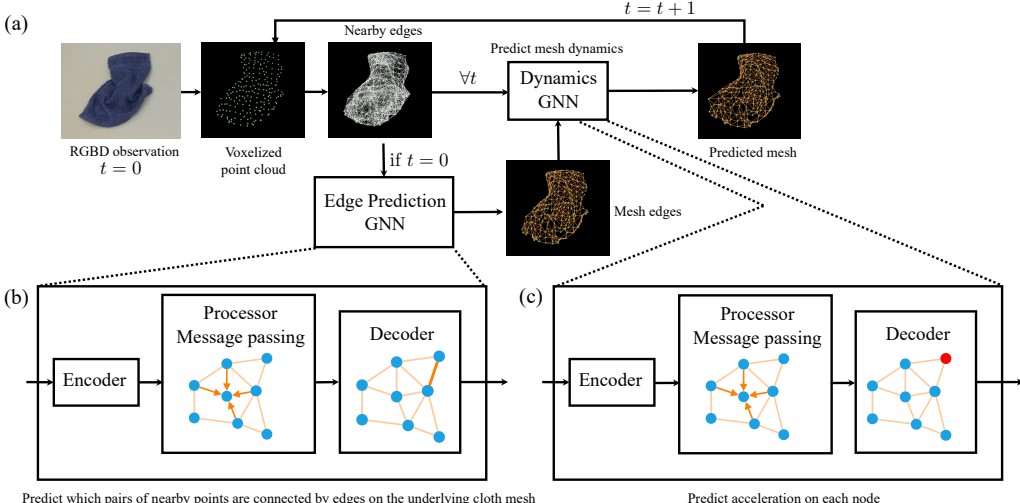

**Figure 2:** (a) Overview of our visible connectivity dynamics model. It takes in the voxelized point cloud, constructs the mesh and predicts the dynamics for the point cloud. (b) Architecture for the edge prediction GNN which takes in the point cloud connected by the nearby edges and predicts for each nearby edge whether it is a mesh edge. (c) Architecture for the dynamics GNN which takes in the point cloud connected by both the nearby edges and the mesh edges and predict the acceleration of each point in the point cloud.

## 3   Method

An overview of our method, VCD (Visible Connectivity Dynamics), can be found in Figure 2. We represent the cloth using a Visible Connectivity Graph, in which we connect points of a partial point cloud with nearby edges and the inferred mesh edges. Next, we learn a dynamics model over this graph, and finally we use this dynamics model for planning robot actions.

### 3.1   Graph Representation of Cloth Dynamics

We represent the state of a cloth with a graph $\langle V, E \rangle$. The nodes $V = \{v_i\}_{i=1\ldots N}$ represent the particles that compose the cloth, where $v_i = (x_i, \dot{x}_i)$ denotes the particle's current position and

velocity, respectively. There are two types of edges $E$ in the graph, representing two types of inter-actions between the particles: mesh edges and nearby edges. The mesh edges, $E^M$, represent the connections among the particles on the underlying cloth mesh. The mesh connectivity is determined by the structure of the cloth and does not change throughout time. Each edge $e_{ij} = (v_i, v_j) \in E^M$ connects nodes $v_i$ to $v_j$ and models the mesh connection between them. The other type of edges are nearby edges, $E^C$, which model the collision dynamics among two particles that are nearby in space. These can be different from the mesh edges due to the folded configuration of the cloth, which can bring two particles close to each other even if they are not connected by a mesh edge. Unlike the mesh edges which stay the same throughout time, these nearby edges are dynamically constructed at each time step based on the following criteria:

$$E_t^C = \{e_{ij} \,|\, ||x_{i,t} - x_{j,t}||_2 < R\}, \tag{1}$$

where $R$ is a distance threshold and $x_{i,t}, x_{j,t}$ are the positions of particles $i, j$ at time step t. Through-out the paper, we use the subscript $t$ to denotes the state of a variable at time step $t$ if the variable changes with time. Additionally, we assume that $E^M \subset E^C$, since a mesh edge connects nodes that are close to each other and hence should also satisfy Eqn. 1.

## 3.2 Inferring Visible Connectivity from a Partial Point Cloud

In the real world, we observe the cloth in the form of a partial point cloud. In this case, we represent the nodes of the graph using the partial point cloud and infer the connectivity among these observed points. We denote the raw point cloud observation as $P_{raw} = \{x_i\}_{i=1..N_{raw}}$, where $x_i$ is the position of each point and $N_{raw}$ is the number of points. We first pre-process the point cloud by filtering it with a voxel grid filter: we overlay a 3d voxel grid over the observed point cloud and then take the centroid of the points inside each voxel to obtain a voxelized point cloud $P = \{x_i\}_{i=1,...,N_p}$. This preprocessing step is done both in simulation training and in the real world, which makes our method agnostic to the density of the observed point cloud and more robust during sim2real transfer.

We create a graph node $v_i$ for each point $x_i$ in the voxelized point cloud $P$. The nearby edges are then constructed by applying the criterion from Eqn. 1. However, inferring the mesh edges is less straightforward, since in the real world we cannot directly perceive the underlying cloth mesh connectivity. To overcome this challenge, we use a graph neural network (GNN) [26] to infer the mesh edges from the voxelized point cloud. Given the positions of the points in $P$, we first construct a graph $\langle P, E^C \rangle$ with only the nearby edges based on Eqn. 1. As we assume $E^M \subset E^C$, we then train a classifier, which is a GNN, to estimate whether each nearby edge $e \in E^C$ is also a mesh edge. We denote this edge GNN as $G_{edge}$. The edge GNN takes as input the graph $\langle P, E^C \rangle$, propagates information along the graph edges in a latent vector space, and finally decodes the latent vectors into a binary prediction for each edge $e \in E^C$ (predicting whether the edge is also a mesh edge). For the edge GNN, we use the network architecture in previous work [7] (referred to as GNS). See Appendix A.1 for the detailed architecture. The edge GNN is trained in simulation, where we obtain labels for the mesh edges based on the ground-truth mesh of the simulated cloth. After training, it can then be deployed in the real world to infer the mesh edges from the point cloud. We defer the description of how we obtain the ground-truth mesh labels in Sec. 3.5.

## 3.3 Modeling Visible Connectivity Dynamics with a GNN

In order to predict the effect of a robot's action on the cloth, we must model the cloth dynamics. While there exists various physics simulators that support simulation of cloth dynamics [27, 28, 29], applying these simulators for a real cloth is still challenging due to two difficulties: first, only a partial point cloud of a crumpled cloth is observed in the real world, usually with many self-occlusions. Second, the estimated mesh edges from Sec. 3.2 may not all be accurate. To handle these challenges, we learn a dynamics model based on the voxelized partial point cloud and its inferred visible connectivity (Sec. 3.2). Formally, given the cloth graph $G_t = \langle V, E \rangle$, a dynamics GNN $G_{dyn}$ predicts the particle accelerations in the next time step, which can then be integrated to update the particle positions and velocities. Here, $V$ refers to the voxelized point cloud, and $E$ refers to inferred visible connectivity that includes both the predicted mesh edges $E^M$ as well as the nearby edges. Our dynamics GNN $G_{dyn}$ uses the similar GNS architecture as the $G_{edge}$. It takes a cloth mesh as input with state information on each node, propagates the information along the graph edges in a latent vector space, and finally decodes the latent vectors into the predicted acceleration on each node. See Appendix A.1 for the detailed architecture of the GNN.

### 3.4 Planning with Pick-and-place Actions

We plan in a high-level, pick-and-place action space over the VCD model. For each action $a = \{x_{pick}, x_{place}\}$, the gripper grasps the cloth at $x_{pick}$, moves to $x_{place}$, and then drops the cloth. As the GNN dynamics model is only trained to predict the changes of the particle states in small time intervals in order to accurately model the interactions among particles, we decompose each high-level action into a sequence of low-level movements, where each low-level movement is a small delta movement of the gripper and can be achieved in a short time. Specifically, we generate a sequence of small delta movements $\Delta x_1, ..., \Delta x_H$ from the high-level action, where $x_{pick} + \sum_{i=1}^{H} \Delta x_i = x_{place}$. Each delta movement $\Delta x_i$ moves the gripper a small distance along the pick-and-place direction and the motion can be predicted by the dynamics GNN in a single step. When the gripper is grasping the cloth, we denote the picked point as $u$. We assume that the picked point is rigidly attached to the gripper; thus, when considering the effect of the $t^{th}$ low-level movement of the robot gripper, we modify the graph by directly setting the picked point $u$'s position $x_{u,t} = x_{pick} + \sum_{i=1}^{t} \Delta x_i$ and velocity $\dot{x}_{u,t} = \Delta x_i / \Delta t$, where $\Delta t$ is the time for one low-level movement step. The dynamics GNN will then propagate the effect of the action along the graph when predicting future states. For the initial steps where the historic velocities are not available, we pad them with zeros for input to the dynamics GNN. If no point is picked, e.g., after the gripper releases the picked point, then the dynamics model is rolled out without manually setting any particle state.

Our goal is to smooth a piece of cloth from a crumpled configuration. To compute the reward $r$ based on either the observed or the predicted point cloud, we treat each point in the point cloud as a sphere with radius $R$ and compute the covered area of these spheres when projected onto the ground plane. Due to computational limitations, we greedily optimize this reward over the predicted states of the point cloud after a one-step high-level pick-and-place action rather than optimizing over a sequence of pick-and-place actions. Given the current voxelized point cloud of a crumpled cloth $P$, we first estimate the mesh edges using the edge predictor $E^M = G_{edge}(\langle P, E^C \rangle)$. We keep the mesh edges fixed throughout the rollout of a pick-and-place trajectory since the structure of the cloth is fixed. In theory, it could be helpful to update the mesh edges based on the newly observed point cloud at each low-level step, but this is challenging due to the heavy occlusion from the robot's arm during the execution of a pick-and-place action. After the execution of each pick-and-place action, new particles may be revealed and we update the mesh edges when re-planning the next action. The pseudocode of the planning procedure can be found in the appendix.

### 3.5 Training in Simulation

The simulator we use for training is Nvidia Flex, a particle-based simulator with position-based dynamics [30, 31], wrapped in SoftGym [28]. In Flex, a cloth is modeled as a grid of particles, with spring connections between particles to model the bending and stretching constraints.

One challenge that we must address is that the points in the observed partial point cloud do not directly correspond to the underlying grid of particles in the cloth simulator. This presents a challenge for obtaining the ground-truth labels used for training the dynamics GNN and the edge GNN, including the acceleration for each point in the observed point cloud and the mesh edges among them. To address this issue, we perform bipartite graph matching to match each point in the voxelized point cloud to a simulated particle by minimizing the Euclidean distance between the matched pairs. Details about the matching can be found in the appendix. After we get the mapping from the points to the simulator particles, the ground-truth acceleration of each point is simply assigned to be the acceleration of its mapped particle, which is used for training the dynamics GNN. For training the edge GNN, a nearby edge is assumed to be a mesh edge if the mapped simulation particles of the edge's both end points are connected by a spring in the simulator.

### 3.6 Graph Imitation Learning for Occlusion Reasoning

To better allow our model to reason about occlusions, we introduce graph-based privileged imitation learning, which transfers features from a teacher model trained on the full cloth to a student model trained on the partial cloth observation. This idea is related to other recent work which trains a student with partial observations to imitate a teacher with full-state information [32, 33, 34]. Specifically, we first train a privileged teacher dynamics model with ground-truth information of the particle state of the full cloth, i.e., it takes as input all particles (including the occluded particles). Next, we train the student model, which takes a partial point cloud as input. The student is trained

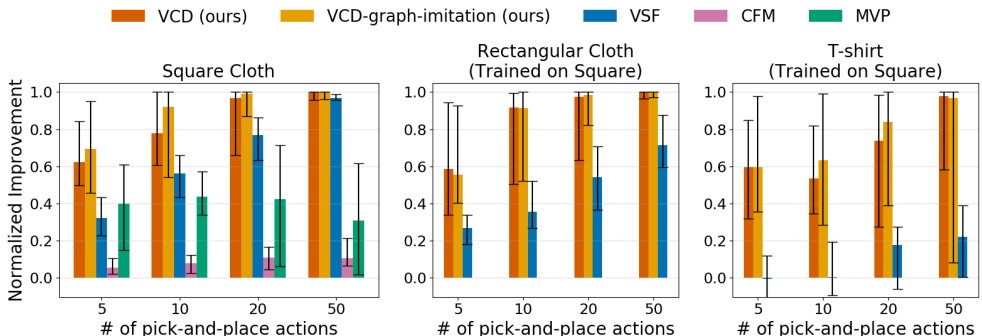

**Figure 3:** Normalized improvement on square cloth (left), rectangular cloth (middle), and t-shirt (right) for varying number of pick-and-place actions. The height of the bars show the median while the error bars show the 25 and 75 percentile. For detailed numbers, see the appendix.

to imitate the features of the corresponding nodes in the teacher [33]. Graph imitation offers direct supervision on the intermediate features and enables the student model to implicitly reason about the occluded part of the cloth, by imitating the features from the teacher which has full state information. We also introduce an auxiliary task of reward (covered area) prediction [35], which implicitly encodes the cloth shape into the learned features. Details on our graph-based privileged imitation learning can be found in the appendix.

## 4 Experiments

### 4.1 Experimental Setup

**Simulation Setup** As mentioned, we use the Nvidia Flex simulator wrapped in SoftGym [28] for training. The robot gripper is modeled as a spherical picker that can move freely in 3D space and can be activated so the nearest particle will be attached to it. For training, we generate random pick-and-place trajectories on a square cloth. The side length of the cloth varies from 25 to 28 cm. For evaluation, we consider three different shapes: 1) the same type of square cloth as used in training; 2) Rectangular cloth, with its length and width sampled from $[19, 21] \times [31, 34]$ cm. 3) Two layered T-shirt (the square cloth used for training was single-layered). For each shape, the experiment was run 40 times, each time with a different initial configuration of the fabric. We report the 25%, 50% and 75% ($Q_{25}, Q_{50}, Q_{75}$) percentiles of the performance. For all our quantitative results, numbers after $\pm$ denotes $\max(|Q_{50} - Q_{25}|, |Q_{75}, -Q_{50}|)$.

Our goal for cloth smoothing is to maximize the covered area of the cloth in the top-down view. We report two performance metrics: Normalized improvement (NI) and normalized coverage (NC). NI computes the increased covered area normalized by the maximum possible improvement $NI = \frac{s-s_0}{s_{max}-s_0}$, where $s_0, s, s_{max}$ are the initial, achieved, and maximum possible covered area of the cloth. Similarly, $NC = \frac{s}{s_{max}}$ computes the achieved covered area normalized by the maximum possible covered area. We report NI in the main paper and NC in the appendix.

We evaluates two variants of our method: Visible Connectivity Dynamics (VCD) and VCD with graph imitation learning. We compare with previous state-of-the art methods for cloth smoothing: VisuoSpatial Foresight (VSF) [5], which learns a visual dynamics model using RGBD data; Contrastive forward model (CFM) [2], which learns a latent dynamics model via contrastive learning; Maximal Value under Placing (MVP) [3], which uses model-free reinforcement learning with a specially designed action space. More implementation details can be found in the appendix.

**Real World Setup** We use our dynamics model trained in simulation to smooth cloth in the real world with a Franka Emika Panda robot arm and a standard panda gripper, with FrankaInterface library [36]. We obtain RGBD images from a side view Azure Kinect camera. We use color thresholding for segmenting the cloth and obtain the cloth point cloud. We evaluate on three pieces of cloth: Two square towels made of cotton and silk respectively, and one t-shirt made of cotton. We use our dynamics model trained in simulation without any fine-tuning. More details are in the appendix.

## 4.2 Simulation Results

For each method, we report the NI after different numbers of pick-and-place actions. A smoothing trajectory ends early when NI>0.95. We note that the edge GNN can achieve a high prediction accuracy of 0.91 on the validation dataset. See appendix for visualizations of the edge GNN prediction.

We first test all methods on the same type of square cloth used in training. The results are shown in Figure 3 (left). Under any given number of pick-and-place actions, VCD greatly outperforms all of the baselines. The graph imitation learning approach described in Section 3.6 further improves the performance. To test the generalization of these methods to novel cloth shapes that are not seen during training, we further evaluate on a rectangular cloth and a t-shirt. For this experiment we only compare VCD to VSF, since VSF achieves the best performance on the square cloth among all the baselines. The results are summarized in Figure 3 (middle and right). VCD shows a larger improvement over VSF on the rectangular cloth. T-shirt is more different from the training square cloth and VSF completely fails, while VCD still shows good generalization. The graph imitation learning still leads to marginal improvement and better stability on rectangular since it has a similar shape to the square cloth. However, as the t-shirt has very different shape compared to the square cloth, VCD-graph-imitation does not lead to much improvement and has larger variance on it.

Since VCD learns a particle-based dynamics model, it incorporates the inductive bias of the cloth structure, which leads to better performance and stronger generalization across cloth shapes, compared to RGB based method like VSF. Please see the appendix for examples of some planned pick-and-place action sequences of our method on all cloth shapes as well as visualizations of the predictions of our model.

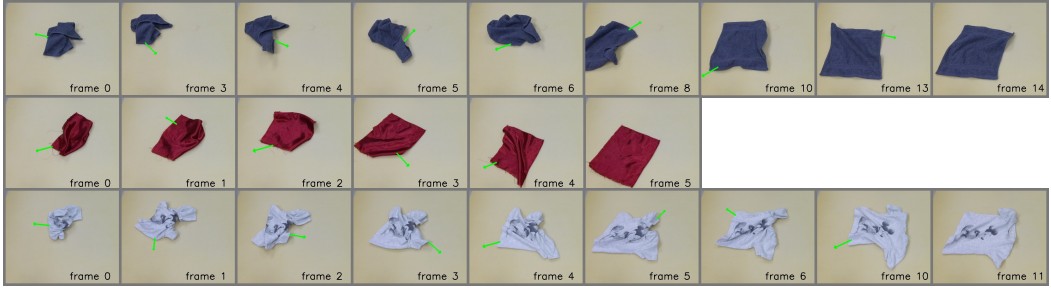

**Figure 4:** Smoothing cloths of different colors, materials and shapes with our method on a Franka robot: square cotton (top), square silk (middle), cotton t-shirt (bottom). Each row shows one trajectory. Frame 0 shows the initial configuration of the cloth, and each frame after shows the observation after some number of pick-and-place actions, with the number labeled on the frame. The green arrow shows the 2D projection of the pick-and-place action executed.

| Material \ # of pick-and-place actions | 5 | 10 | 20 | Best |
|---|---|---|---|---|
| Cotton Square Cloth | $0.342 \pm 0.265$ | $0.725 \pm 0.445$ | $0.941 \pm 0.360$ | $0.941 \pm 0.153$ |
| Silk Square Cloth | $0.456 \pm 0.197$ | $0.643 \pm 0.391$ | $0.952 \pm 0.229$ | $0.952 \pm 0.095$ |
| Cotton T-Shirt | $0.265 \pm 0.119$ | $0.356 \pm 0.096$ | $0.502 \pm 0.135$ | $0.619 \pm 0.155$ |

**Table 1:** Normalized improvement of VCD in the real world.

## 4.3 Real-world Results

We also evaluate our method for smoothing in the real world. We only evaluate VCD (i.e., without graph imitation learning) since it works more stably in simulation. Unfortunately, we were not able to evaluate the baselines in the real world due to the difficulties of transferring their RGB-based policies from simulation. All of the baselines use RGB data as direct input to the dynamics model or the learned policy, making them sensitive to the camera view and visual features. In contrast, our method uses a point cloud as input, which makes it robust to the camera position as well as variation in visual features such as the cloth color or patterns. The point cloud representation allows our method to easily transfer to the real world.

We evaluate 12 trajectories for each cloth. The quantitative results are in Table 1 and a visualization of smoothing sequences is shown in Figure 4. Despite the drastic differences of the cotton and silk cloths in visual appearances, shapes, as well as the different dynamics, our model is able to smooth the cotton and silk cloths and generalize well to t-shirt. We also report the performance if our method is able to terminate optimally in hindsight and choose the frame with the highest performance in each trajectory; the result is shown in the last column of Table 1. Videos of complete trajectories and the model predicted rollouts can be found on our project website.

### 4.4 Ablation Studies

We perform the following ablations to study the contribution of each component of our method. The first ablation replaces the learned GNN dynamics model with the Flex simulator to test whether a learned dynamics model performs better for our task than the physical simulator. In more detail, after we use the edge GNN to infer the mesh edges on the point cloud, we create a cloth using Flex where a particle is created at each location of the voxelized points and a

| Algorithm | Normalized Improvement |
|---|---|
| VCD (Our method) | **0.778 ± 0.206** |
| Replace dynamics GNN with Flex | 0.616 ± 0.143 |
| No edge GNN (dynamic nearby edges) | 0.531 ± 0.298 |
| No edge GNN (fixed nearby edges) | 0.599 ± 0.327 |
| Remove edge GNN at test time | 0.259 ± 0.118 |

**Table 2:** Normalized improvement of all ablations in simulation after 10 pick-and-place actions.

spring connection is added for each inferred mesh edge. The results is shown in Table 2, row 2. We see that using the Flex simulator instead of the dynamics GNN produces significantly worse performance. The main reason is that the cloth created from the partial point cloud with the inferred mesh edges deviates from the common cloth mesh structure used in Flex; thus, using the Flex simulator under this condition does not create realistic dynamics. On the other hand, the dynamics GNN is trained directly on the partial point cloud; therefore it can learn to compensate for the partial observability when predicting the cloth dynamics. This ablation validates the importance of using a dynamics GNN to learn the dynamics of the partially observable point cloud.

The next set of ablations aims to test whether using an edge GNN to infer the mesh edges as described in Section 3.1 is necessary for learning a good dynamics model. First, we train a dynamics GNN without using the edge GNN, where the edges are constructed solely based on distance by Eqn. (1). Since this ablation does not use an edge GNN, it cannot have two different edge types (nearby edges vs mesh edges). Thus at test time, all edges can either be kept fixed throughout the trajectory (similar to the mesh edges in our model), or dynamically reconstructed using Eqn. (1) at each time step (similar to the nearby edges in our model). The results of these two ablations are shown in Table 2, rows 3 and 4. As can be seen, the performance is worse without the edge GNN.

Additionally, we perform another ablation where we train with both nearby edges and mesh edges, but at test time, we do not use an edge GNN to infer the edge type; instead we consider the edges that satisfy the criteria of Eqn. (1) in the first time step as the mesh edges. The result of this ablation is shown in Table 2, row 5. The performance is again much worse. All these ablations validate the importance of using an edge GNN to infer the mesh edges.

## 5   Conclusion

In this paper, we propose the visible connectivity dynamics (VCD) model, that infers a visibility connectivity graph from the partial point cloud and learns a particle-based dynamics model over the graph for planning to perform cloth smoothing. VCD has the advantage of posing strong inductive bias that fits the underlying cloth physics, being invariant to visual features, and being interpretable. We show that VCD greatly outperforms previous state-of-the-art methods for cloth smoothing, and achieves zero-shot sim-to-real transfer on a Franka arm for smoothing various types of cloth.

Our work demonstrates the importance of the choice of state representation for efficient and generalizable manipulation, as well as the benefits of a graph-based representation. While there may not be a universal representation suitable for all objects, we believe that a graph representation can be an important alternative to raw images or latent vectors, especially for deformable objects such as cloth. In the future, we hope VCD can also be applied to different types of object manipulation tasks, such as manipulation of cables, bags, and food.

**Acknowledgments**

We would like to thank Wen-Hsuan Chu for his initial Pytorch implementation of the GNS model. We would like to thank members of the RPAD lab, Gokul Swamy and Erica Weng for their feedback on the early draft of the paper. This material is based upon work supported by the National Science Foundation under Grant No. IIS-1849154 and LG Electronics.

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
