# OpenReview forum: "Learning Visible Connectivity Dynamics for Cloth Smoothing"
_robot-learning.org/CoRL/2021/Conference — CoRL2021 Poster_

### Official Review · Reviewer_9vXf · 2021-07-22

**Originality:** Very Good
**Technical Quality:** Good
**Clarity Of Presentation:** Good
**Impact:** 4

**Recommendation:**

Strong Accept: I recommend accepting the paper and will argue for my recommendation even if other reviewers hold a different opinion.

**Summary:**

This paper proposed a general and efficient cloth smoothing scheme using the predicted visible connectivity dynamics (VCD).
By representing the cloth shape as a graph structure using GNN with point cloud data as input, the authors can obtain a representation of the dynamics that is robust to the effects of visibility and self occlusion.
Although the control policy used seems to be simple, it is still able to achieve smoothing of multiple types of cloths efficiently.

**Issues:**

- The proposed graph representation method seems to produce dense graphs, but is there any problem of computational cost?
Also, isn't there a trade-off between computational cost and accuracy in terms of the degree of voxelization?
- As auxiliary purpose inspired by active sensing to the control strategy, is it possible to give the reward for behavior that increases the prediction accuracy of VCD?
- It seems to be a learning problem without accurate supervised signals.
I would like to know whether this learning problem is unstable due to noises and/or outliers from wrong annotations.


**Reviewer Expertise:**

Good: General knowledge of the area

**Strengths And Weaknesses:**

Strengths:
- The effectiveness of the framework was clearly demonstrated through careful comparative experiments.
- The contribution of each component in the framework was also verified by conducting ablation experiments, which increased the value of this paper.
- I think this paper is not on a new technology, but rather on a proper system integration for using GNNs for cloth shape prediction with reliable procedure.

Weaknesses:
- It is unclear how to optimize the design parameters that may be correlated with the size of the object, such as voxelization and node connection threshold.
- The control policy was simple and did not make full use of the high-level dynamics model obtained, and although it was faster than conventional methods, it was still not at a practical level.
- It is difficult to say that this paper is self-contained, since much of the information is referred to in appendices and the past papers.


**Summary Of Recommendation:**

The proposed VCD has been carefully implemented from the prediction method to verification, and is a significant contribution.
Although there are still some open problems, they are not within the scope of this paper and are well worthy of acceptance.

---

> ### Author Response · Authors · 2021-08-30
> **Response to Reviewer 9vXf**
>
> Thank you for your thoughtful review and the positive feedback. We have updated our paper and the appendix (highlighted in red) and we address each of your concern below:
>
> > It is unclear how to optimize the design parameters that may be correlated with the size of the object, such as voxelization and node connection threshold.
>
> We use the same parameters (including voxelization and node connection threshold R) for all of our simulation and real-world experiments and on objects with different sizes (square cloth, rectangular cloth, and t-shirt). This demonstrates that these parameters are robust to the size and shape of the object. Below we explain how these parameters were chosen. We have also updated the Supplementary Table 1 of Appendix A.2 on these design choices.
>
> R is related to the voxel size. We choose R such that neighboring points on the cloth are connected after voxelization; hence, we choose R to be twice the voxel size (e.g. voxel size = 2.16cm, R = 4.5cm).
> The voxel size is chosen to balance the accuracy of the particle representation and the computational efficiency:
> If voxelization is too small, then there will be a large number of particles, resulting in slow inference
> If voxelization is too large, then points on the cloth will become too sparse, losing the benefit of a particle representation. In our experiments, we set the voxelization size to be 3 times the particle size in Flex, although we did not experiment with this parameter.
>
> > The control policy was simple and did not make full use of the high-level dynamics model obtained,
>
> We view the control policy to be orthogonal to the main contribution of our paper, which is about building a graph dynamics model for a cloth under partial observability. We found that our simple control policy was still able to significantly outperform the baselines on this task, due to our graph dynamics model.  We leave better control policies to future work.
>
> > and although it was faster than conventional methods, it was still not at a practical level.
>
> It would be an interesting future direction to combine our dynamics model with a policy (e.g. similar to MBPO (Janner et al. 2019) or other papers that combine model-based and model-free methods for policy learning) for faster inference.
>
> > It is difficult to say that this paper is self-contained, since much of the information is referred to in appendices and the past papers.
>
> The details of our GNN architecture is described in Appendix A.1 and we have also updated Section 3.2 and 3.3 of the paper with a high-level description of the GNN. Unfortunately, due to space constraints, we could not explain all of the GNN architectural details that were largely taken from prior work within the main text; instead, we focused on explaining the main contribution of our approach.
>
> > The proposed graph representation method seems to produce dense graphs, but is there any problem of computational cost? Also, isn't there a trade-off between computational cost and accuracy in terms of the degree of voxelization?
>
> After voxelization, we typically have 100-200 nodes in the graph for the experiments we have performed, which is not considered a very large graph compared to other graph neural networks in the literature. For example, in Sanchez-Gonzalez et al [7], the number of nodes in the graphs range from 1k to 20k. We agree that there is a trade-off between computational cost and accuracy, as described in response to your previous question about voxelization parameters.
>
> > As auxiliary purpose inspired by active sensing to the control strategy, is it possible to give the reward for behavior that increases the prediction accuracy of VCD?
>
> This is an interesting direction. One interesting future work can be finding actions that reveal the occluded part of the cloth.
>
> > It seems to be a learning problem without accurate supervised signals. I would like to know whether this learning problem is unstable due to noises and/or outliers from wrong annotations.
>
> We currently train in simulation and transfer to the real world. Thus during training, we have access to the ground-truth acceleration of each point in the point cloud in the simulator. However, if we want to train in the real world, then we will need to track the point cloud to estimate the acceleration of each point; in such a case, our method would need to deal with inaccuracy in tracking. We leave such an extension to future work.

---

> > ### Comment · Reviewer_9vXf · 2021-09-02
> > **Thank you for your response**
> >
> > I thank the authors for addressing my comments. I do not change my evaluation as I had already accepted it, but I am looking forward to seeing the sophisticated application of the obtained model.

---

### Official Review · Reviewer_hhGm · 2021-07-23

**Originality:** Excellent
**Technical Quality:** Excellent
**Clarity Of Presentation:** Excellent
**Impact:** 4

**Recommendation:**

Strong Accept: I recommend accepting the paper and will argue for my recommendation even if other reviewers hold a different opinion.

**Summary:**

Authors proposed a particle-based model to enable manipulation with cloths. The proposed approach, VCD, is able to learn effective models of the cloth, and it works effectively under the condition of partial observability. The efficacy of the approach is validated both in simulation and on a real robot.

**Issues:**

- I mainly have one concern about the approach on real robots: In my understanding, the number of nodes are quite dense for VCD. In the experiment, is the graph generation robust enough to the depth sensor noise? I think authors did not mention this part in the paper. I would like to see the factor of depth sensor noise discussed to fully understand its strength and weakness in the real robot scenario.


**Reviewer Expertise:**

Very good: Comprehensive knowledge of the area

**Strengths And Weaknesses:**

Strength:
A very interesting way to model the cloth with particles
A very nice way to use teacher-student model to learn a model for partial observability
Impressive results that the learned model can sim-to-real transfer

Weakness:
The approach is only tested on the smoothing task. It is unclear at this point how the approach performs on other tasks such as folding.
It is hard to specify specific goal configurations in the current formulation.
The current formulation can only consider a single deformable object at a time.


**Summary Of Recommendation:**

I find this paper very easy to follow even without personal experience working with deformable object manipulation. Authors also make their contribution clear in the writing. Experiments are very extensive and results are very impressive.

---

> ### Author Response · Authors · 2021-08-30
> **Response to Reviewer hhGm**
>
> Thank you for your thoughtful review and the positive feedback. We have updated our paper and the appendix (highlighted in red) and we address each of your concern below:
>
> > The approach is only tested on the smoothing task. It is unclear at this point how the approach performs on other tasks such as folding.
>
> We have added experiments of using VCD for cloth folding to Section E of the updated appendix, with results in Supplementary Table 5 and Supplementary Figures 10, 11, and 12. We test VCD for 3 different goal folding configurations (specified as images/point clouds), and with 3 different cost functions (a groundtruth cost using particles, chamfer distance between achieved and target point clouds, and IOU between achieved and target depth images) for planning.
> The experiment results show that VCD performs fairly well for the folding task. Please see Section E of the updated appendix for detailed experiment descriptions, and the quantitative/qualitative results.
>
> > I mainly have one concern about the approach on real robots: In my understanding, the number of nodes are quite dense for VCD. In the experiment, is the graph generation robust enough to the depth sensor noise?
>
> We have added an experiment to the updated appendix in Section F to test the robustness of VCD to the depth sensor noise. Specifically, we evaluate VCD with different levels of noises injected to the depth map in simulation. Our results demonstrate that VCD is robust to the noise level of the Azure Kinect depth camera, which is the camera we use for the real-word experiment. Please see Section F of the updated appendix  for detailed experiment descriptions and results.

---

> > ### Comment · Reviewer_hhGm · 2021-09-03
> > **Final comment**
> >
> > I would like to thanks the authors for the response. It's good to see additional experiments on folding, although it is still in simulation. It would be nice to see cloth folding on real robots in the future as well.

---

### Official Review · Reviewer_Lzhn · 2021-07-28

**Originality:** Good
**Technical Quality:** Good
**Clarity Of Presentation:** Good
**Impact:** 3

**Recommendation:**

Weak Accept: I recommend accepting the paper, but will not argue for my recommendation if the majority of other reviewers have a different opinion.

**Summary:**

In this work, a method that learns a visible connectivity dynamics model is presented for learning the observable portion of the point cloud and uses that for cloth smoothing task. The results of the proposed method was compared with state-of-the-art model-based and model-free reinforcement learning methods, showing promising results. The experiments also showed that their method is able to generalise for different shapes of fabrics and demonstrated zero-short sim-to-real transfer.

**Issues:**

1.	In Figure 3 there is no definition of error bars, please define their meaning (are these standard deviation, standard error or the percentil range described at the beginning of section 4.1.).
2.	The article is well organized and easy to read, however, in my opinion, the conclusion section could have been elaborated a bit more.
3.	The statement “for all shapes, we test on 40 initial configurations” (line 238) seems not fully clear for me, this could be rephrased. I believe authors mean that for each shape the experiment was run 40 times, each time with different initial configuration of the fabric (cloth or T-shirt).


**Reviewer Expertise:**

Very good: Comprehensive knowledge of the area

**Strengths And Weaknesses:**

Strengths:
1. The main contribution of the paper is implementation of a particle-based dynamics model for visible portion of cloth point cloud (visible connectivity dynamics) to accomodate a partial visibility. The model learns dynamics using the network presented in [7].
2. The shape specific training is addressed by having a student model with partial observations learns to imitate a teacher model with full-state information.

Weaknesses:
1. The proposed model predicts the mesh of the fabric but it is not clear that how robot action is taken into consideration in the model. It is obvious that the robot motion, even small motions, will change the state of the fabric and as a result will change the mesh to be predicted. In line 11 of that algorithm, it is mentioned that the inputs to the dynamic network are: the current cloth points (after the pick points were moved), historical cloth points, picked point and set of collision and mesh edges. It is surprising that the robot action is not taken in the input as it will change the state of the fabric and it is also easy to obtain.
2. The presentation can be further improved by including more details of the presented method. For example, how is the distance threshold R determined? I believe the choice is important in determining the collision edges and R may be different for different cloth types and sizes.

**Summary Of Recommendation:**

As written in the Strengths and Weaknesses section, the paper proposes an interesting method for cloth smoothing. The proposed approach is leveraging a network presented in previous works, but implemented in a different application, which could advance the development of methods in robot cloth smoothing. The paper is well presented in general but the presentation can be further improved by including more details of the presented method. For example, how is the distance threshold R determined?

---

> ### Author Response · Authors · 2021-08-30
> **Response to Reviewer Lzhn**
>
> Thank you for your constructive review. We have updated our paper and the appendix (highlighted in red) and we address each of your concern below:
>
> > it is not clear that how robot action is taken into consideration in the model. In line 11 of that algorithm, it is mentioned that the inputs to the dynamic network are: the current cloth points (after the pick points were moved), historical cloth points, picked point and set of collision and mesh edges. It is surprising that the robot action is not taken in the input as it will change the state of the fabric and it is also easy to obtain.
>
> Our dynamics model considers the effect of the robot action by directly changing the position and the velocity of the point that is picked by the robot gripper; this updated position and velocity is input to the dynamics model. This is described in Section 3.4 and we have updated the corresponding description, as well as the description in Appendix A.1, to make this more clear.
>
> This parameterization of the action space is different from a typical MLP latent vector dynamics model where the action vector is directly concatenated with a latent vector of the state.  With our action parameterization, the robot action explicitly affects the state of the grasped point on the cloth, which can make the dynamics model more sample-efficient to learn and more generalizable. This action parameterization is enabled by our use of the particle representation of the cloth model.
>
> > how is the distance threshold R determined? I believe the choice is important in determining the collision edges and R may be different for different cloth types and sizes.
>
> R is related to the voxel size. We choose R such that neighboring points on the cloth are connected after voxelization; hence, we choose R to be twice the voxel size (e.g. voxel size = 2.16cm, R = 4.5cm). The voxel size is chosen to balance the accuracy of the particle representation and the computational efficiency: If voxelization is too small, then there will be a large number of particles, resulting in slow inference; If voxelization is too large, then points on the cloth will become too sparse, losing the benefit of a particle representation.
> In our experiments, we set the voxelization size to be 3 times the particle size in Flex, although we did not experiment with this parameter.
>
> We have added a summary table of the hyperparameter values in Supplementary Table 1.
>
> > In Figure 3 there is no definition of error bars, please define their meaning.
>
> The error bars refer to the 25 and 75 percentile. We have updated the caption of Figure 3 to clarify this.
>
> > The article is well organized and easy to read, however, in my opinion, the conclusion section could have been elaborated a bit more.
>
> We have updated the conclusion in the paper.
>
> > The statement “for all shapes, we test on 40 initial configurations” (line 238) seems not fully clear for me, this could be rephrased.
>
> We have updated Section 4.1 to clarify this point.

---

> > ### Comment · Reviewer_Lzhn · 2021-09-03
> > **Thanks for the response**
> >
> > I thank the authors to take time and effort in the responding to my comments. I would love to see how the complicated models work on the real robots. It would also be interesting to see more results if robot action is taken into consideration directly (as it is more natural from the robot perspective). I will keep my positive rating.

---

### Official Review · Reviewer_ViDy · 2021-07-30

**Originality:** Very Good
**Technical Quality:** Very Good
**Clarity Of Presentation:** Good
**Impact:** 4

**Recommendation:**

Weak Accept: I recommend accepting the paper, but will not argue for my recommendation if the majority of other reviewers have a different opinion.

**Summary:**

This paper presents a new model-based RL approach for cloth manipulation, which involves learning a novel graph-neural-net based dynamics of point-clouds. The system appears to outperform prior model-free and model-based approaches, and shows impressive sim2real transfer.

**Issues:**

See weaknesses above.

**Reviewer Expertise:**

Very good: Comprehensive knowledge of the area

**Strengths And Weaknesses:**

Strengths

- The paper proposes a viable solution to the Cloth manipulation problem, which is very hard because of partial observability, high configuration space, and underlying real-world soft-body dynamics.

- Zero shot sim2real transfer is impressive.

Weaknesses

- I found the paper hard to read in certain sections particularly the details of GNNs and how exactly the dynamics is simulated.

- Not much analyses is provided on how sensitive the overall success is to the choice of architecture

- It is surprising that one step dynamics fitting works for this application since other model-based RL approaches crucially rely on multi-step losses or DAGGER like corrections.

- One baseline that would be interesting is blackbox policy optimization against FLEX cloth models directly.


**Summary Of Recommendation:**

Novel approach for model-based cloth manipulation that shows promising sim2real transfer and relies on a new way to use GNNs to model point cloud dynamics. Overall, the presentation of architectural details could be significantly improved.

---

> ### Author Response · Authors · 2021-08-30
> **Response to Reviewer ViDy**
>
> Thank you for your thoughtful review. We have updated our paper and the appendix (updates are written in red font) and we address each of your concerns below:
>
> > I found the paper hard to read in certain sections particularly the details of GNNs and how exactly the dynamics is simulated.
>
> We have updated our paper in Sections 3.2 and 3.3, with a high-level description of our GNNs and the full details of our GNN architecture is described in Appendix A.1.
>
> > Not much analyses is provided on how sensitive the overall success is to the choice of architecture
>
> The specific choice of GNN architecture is not the focus of our paper so we directly adopt the GNS architecture from Sanchez-Gonzalez et al [7] . We modified the GNS architecture to include a global model, as was done previously in Battaglia, Peter W et al [26]; we have added experiments to show that adding a global model yields improved performance (see ablation in Supplementary Figure 15 (left) and Appendix H). We also conduct a sensitivity analysis on the number of message passing steps in the dynamics GNN, and the results show that our model is generally robust to this parameter (see Supplementary Figure 15 (right) and Appendix H). For other parameters, a comprehensive analysis of the GNS architecture can be found in Sanchez-Gonzalez et al [7] at Figure 4 and Figure C.1 of that paper. Overall, we consider these specific architectural details to be orthogonal to our main contribution.
>
> > It is surprising that one step dynamics fitting works for this application since other model-based RL approaches crucially rely on multi-step losses or DAGGER-like corrections.
>
> This is a good point.  Our thoughts on this question are as follows:
> 1. We train from a diverse set of initial states with random actions, so our training data covers a large part of the state space.
> 2. As our experiments show, our dynamics model is very general due to the inductive bias of GNN (it generalizes zero-shot to rectangular cloths and t-shirts) and thus it likely generalizes to out-of-distribution states as well, minimizing the need for DAgger-like corrections.
> 3. We might get even better performance with multi-step losses, but we found that we could achieve fairly good performance even without them. The idea of using multi-step losses is fairly orthogonal to the main contribution of our paper.
>
> > One baseline that would be interesting is blackbox policy optimization against FLEX cloth models directly.
>
> We have added Supplementary Figure 14 in Section G of the updated appendix that conducts a comparison with an oracle which uses the ground-truth FleX cloth model in the simulator for planning. The oracle uses the same planning method as VCD and achieves almost the maximum possible performance. This shows that better performance can be achieved if the full cloth model and dynamics can be better estimated, which we leave for future work.

---

### Author Response · Authors · 2021-08-31
**Response to all**

We thank all the reviewers for the thoughtful reviews and the positive feedback. We have updated our paper and the appendix (updates are written highlighted in red font). We have also done additional experiments as suggested by the reviewers. Specifically, we tested our method on cloth folding, where the goal configurations are specified via depth images or point clouds, and found it to perform well.

***All additional experimental results are summarized in the updated appendix in Section E (VCD for folding), Section F (robustness of VCD to depth sensor noise), Section G (comparison to an oracle dynamics model) and Section H (Analysis on GNN architecture choice).***

---

### Meta-Review · Area_Chair_Zkrp · 2021-08-13

**Recommendation:** Accept (Poster)
**Confidence:** 5

**Metareview:**

All reviewers were generally positive, and found the use of graph neural networks as a dynamics model for deformable objects, to be an interesting and novel contribution. Although there were no major weaknesses raised, in the rebuttal, authors should address the queries raised by the reviewers in order to maintain good review scores. In particular, I would like to hear the authors' opinions on how this method would work with tasks other than smoothing, such as how goal configurations would be specified more generally, as raised by Reviewer hhGm.

--------

The authors have provided an updated paper with new experiments, and have addressed many of the issues raised by the reviewers. This is a novel method for deformable object manipulation, with some good real-world results. All four reviewers recommend acceptance of the paper. Now that the authors have shown that the method works for folding as well as smoothing, I recommend updating the title of the paper.

---

> ### Author Response · Authors · 2021-08-30
> **Response to all**
>
> We thank all the reviewers for the thoughtful reviews and the positive feedback. We have updated our paper and the appendix (updates are written highlighted in red font). We have also done additional experiments as suggested by the reviewers. Specifically, we tested our method on cloth folding, where the goal configurations are specified via depth images or point clouds, and found it to perform well.
>
> ***All additional experimental results are summarized in the updated appendix in Section E (VCD for folding), Section F (robustness of VCD to depth sensor noise), Section G (comparison to an oracle dynamics model) and Section H (Analysis on GNN architecture choice).***

---

### Decision · Program_Chairs · 2021-09-13

**Decision:**

Accept (Poster)

**Comment:**

All reviewers were generally positive, and found the use of graph neural networks as a dynamics model for deformable objects, to be an interesting and novel contribution. Although there were no major weaknesses raised, in the rebuttal, authors should address the queries raised by the reviewers in order to maintain good review scores. In particular, I would like to hear the authors' opinions on how this method would work with tasks other than smoothing, such as how goal configurations would be specified more generally, as raised by Reviewer hhGm.

--------

The authors have provided an updated paper with new experiments, and have addressed many of the issues raised by the reviewers. This is a novel method for deformable object manipulation, with some good real-world results. All four reviewers recommend acceptance of the paper. Now that the authors have shown that the method works for folding as well as smoothing, I recommend updating the title of the paper.